# LTBP4 in Health and Disease

**DOI:** 10.3390/genes12060795

**Published:** 2021-05-23

**Authors:** Chi-Ting Su, Zsolt Urban

**Affiliations:** 1Department of Internal Medicine, Renal Division, National Taiwan University Hospital Yunlin Branch, Douliu 640, Taiwan; chitingsu@gmail.com; 2Department of Human Genetics, Graduate School of Public Health, University of Pittsburgh, Pittsburgh, PA 15261, USA; 3Department of Medicine, National Taiwan University Cancer Center Hospital, Taipei 106, Taiwan

**Keywords:** LTBP4, TGFβ, elastogenesis, medicine

## Abstract

Latent transforming growth factor β (TGFβ)-binding protein (LTBP) 4, a member of the LTBP family, shows structural homology with fibrillins. Both these protein types are characterized by calcium-binding epidermal growth factor-like repeats interspersed with 8-cysteine domains. Based on its domain composition and distribution, LTBP4 is thought to adopt an extended structure, facilitating the linear deposition of tropoelastin onto microfibrils. In humans, mutations in LTBP4 result in autosomal recessive cutis laxa type 1C, characterized by redundant skin, pulmonary emphysema, and valvular heart disease. LTBP4 is an essential regulator of TGFβ signaling and is related to development, immunity, injury repair, and diseases, playing a central role in regulating inflammation, fibrosis, and cancer progression. In this review, we focus on medical disorders or diseases that may be manipulated by LTBP4 in order to enhance the understanding of this protein.

## 1. Introduction

Latent transforming growth factor β-binding proteins (LTBPs) serve as mediators to organize elements for matrix microfibril bundles and regulate the signaling of transforming growth factor β (TGFβ). They play a crucial role in matrix homeostasis and are required for elastogenesis. LTBP4, a member of the LTBP family, contains numerous modular domains, including extended calcium-binding epidermal growth factor-like repeats interspersed with 8-cysteine (TB and hybrid) domains [1,2]. LTBP4 binds covalently to TGFβ1 and is required for the correct folding of the latent TGFβ1 complex. LTBP4–TGFβ1 complexes are secreted from the cytoplasm and are incorporated into the matrix; this binding is essential for TGFβ1 activation [3,4]. TGFβ1 is a pleiotropic cytokine that is thought to play an essential role in chronic fibrosis in many organs, including pulmonary, kidney, liver, and skin tissues. In addition, in cancer, desmoplasia of the extracellular matrix (ECM), angiogenesis, inflammation, and reduced anti-carcinogenic adaptive immune cellular responses in the tumor microenvironment may induce carcinogenesis [5,6].

Although there are a few comprehensive review articles on the LTBP family, most of these articles address the biological functions of LTBPs associated with TGFβ regulation, elastic fiber assembly, microfibril organization, and structural complexity. In this review, we describe the roles of LTPB4 in various diseases and its potential implications in medicine.

## 2. Biological Implications

### 2.1. LTBP4 Structure and Isoforms

LTBP4 is a secreted ECM glycoprotein with structural homology to fibrillins and belongs to a family comprising four LTBPs. This matrix protein has multiple calcium-binding epidermal growth factor-like domains and contains a domain unique to LTBP4 and fibrillins: the eight cysteine (8-Cys) or TGFβ-binding (TB) domain, which has four intradomain cysteine residues [7,8,9]. LTBP4 covalently binds the small latent complex of TGFβ and deposits the complex into the ECM after secretion [10], and binds only TGFβ1 [11]. LTBP4 forms a large latent complex (LLC) by binding to the mature dimer of TGFβ, TGFβ propeptide dimer (also referred to as latency-associated peptide (LAP)), and a single-molecule LTBP4 [12]. To exert its signaling and activation effects, TGFβ must dissociate from its propeptide. The complex is secreted from cells, during which the TGFβ dimer remains associated with mature TGFβ via non-covalent binding.

In mammalian cells, two major isoforms of LTBP4, known as the long form (LTBP4L; human: NM_001042544.2; murine: NM_175641.2) and short form (NM_001042545.2; murine: NM_001113549), have been identified. These LTBP4 isoforms are encoded by two variants with different N-terminal splice sites that are independently expressed from separate promoters with identical sequences in the central fragments and carboxy-terminal regions [8,13]. This expression leads to distinct deposition patterns in the ECM in a tissue-specific manner.

### 2.2. TGFβ-Related Signaling

LTBP–TGFβ complexes are required for LTBP to incorporate into the ECM. If the TGFβ–LLC complex does not incorporate into the ECM, the levels of active TGFβ may be reduced. LTBP4 has been reported to enable latent TGFβ secretion and regulate latent TGFβ activation [12]. TGFβ is a potent pleiotropic cytokine, the synthesis and activation of which are associated with LTBPs. Sequestration of the latent form of the TGFβ complex in the ECM is important for proper TGFβ activation. Although incorporated into the matrix in the form of LLC, TGFβ remains inert. Activation of latent TGFβ in different tissues is initiated by several biological processes, such as hypoxia and stretch, dependent on TGFβ signaling, thereby leading to the release of TGFβ from the complex to bind to its specific TGFβ receptors.

Once TGFβ is activated from its propeptide dimer form, it binds to a heteromeric transmembrane TGFβ receptor complex, formed by the dimeric TGFβ1 receptor Alk5 (also known as TβRI) and the dimeric TGFβ type II receptor [14,15]. For downstream signaling, the canonical TGFβ family ligand signaling begins with phosphorylation of TβRI on serine residues of human receptor-regulated/activated (R-) Smad proteins. Smad1, Smad5, and Smad8 are phosphorylated by Alk1- and Alk2-containing receptors, whereas Smad2 and Smad3 are phosphorylated by Alk5/TβRI in the TGFβ receptor. In contrast, the two inhibitory (I-) Smads, Smad6 and Smad7, provide negative feedback control of canonical signaling [16]. Inhibitory Smad7 antagonizes TGFβ signaling by inhibiting R-Smad binding to TβRI [17]. Moreover, the activated Smad2/3 complex cooperates with co-Smad4 and translocates into the nucleus, leading to increased gene expression regulated by co-activators and cell-specific transcription factors. In addition to canonical Smad signaling, TGFβ can activate various other pathways, including JNK, p38, ERK, MAPK, and Rho/ROCK, to influence multiple cellular signaling systems such as the actin cytoskeleton, tight-junction strands, and transcriptional regulation [18]. TGFβ signaling is part of a complex network of intracellular signaling, with crucial connection points in development, immunity, and malignancy [19]. Smad complexes bind DNA with low affinity and act in connection with other transcription factors that define binding sites and repressor/activator activities [20,21,22].

Because of its involvement in context-dependent signaling pathways, TGFβ1 can act in opposing manners, which includes its impact on different cell types even in similar environments. Such context-dependent signaling (i.e., dependent on the signaling state of the target cell) is a factor affecting the opposing effects of TGFβ1 on different cells in the same environment [17]. The same cancer cells inhibited by TGFβ1 to reduce their growth and metastatic potential in the early phases of tumor development progressively escape this suppressive effect and use growth factors to drive metastasis [23].

### 2.3. Non-TGFβ-Dependent Functions

The precise molecular mechanism underlying LTBP4 regulation of elastic fiber assembly, elastogenesis, is unclear. Studies have shown that LTBP4 plays an essential role in regulating elastic fiber assembly and binds fibulin-5 to help assemble microfibrils. Recombinant LTBP4 promotes elastogenesis in human dermal fibroblasts and interacts with fibrillin [24,25], and fibulin-5 may bind to fibrillin independently of LTBP4. Furthermore, Fbln5^−/−^; LTBP4S^−/−^ double-knockout mice present more severe defects than LTBP4S^−/−^ mice, including defective alveolar septations and lung features resembling those of Fbln5^−/−^ mice [26]. When LTBP4 is deficient, abnormally aggregated elastin–fibulin-5 complexes bind to the microfibril, which is more disruptive to lung development than the incorporation of elastin in the absence of both LTBP4 and fibulin-5. Bultmann-Mellin et al. reported that Ltbp4 is essential for the incorporation of fibulin-4 in the ECM [27]. For proper linear deposition on the microfibril network, the interaction of LTBP4 with a fibulin-5–fibulin-4–tropoelastin complex seems to be critical. A summary graph in Figure 1 illustrates the potential matrix proteins that have been reported to interact with LTBP4.

### 2.4. LTBP4 versus Other LTBP Family Members

The LTBP isoforms LTBP1, LTBP2, LTBP3, and LTBP4 are important ECM proteins that belong to the fibrillin superfamily. Fibrillins (~340 kDa) are larger than LTBPs, which have an average molecular mass of 120–200 kDa. LTBP1, LTBP3, and LTBP4 covalently bind to latent TGFβ and have different abilities to incorporate into large fibrillary structures in the ECM [2,28]. ECM proteins interact with TGFβ family ligands, enabling them to act as mediators by binding the mature growth factor to their propeptic portion. Although most TGFβ family members non-covalently aggregate into ECM components, the three TGFβ isoforms show covalent binding to LTBP in the ECM strongly [29]. However, in the LTBP family, LTBP2 is missing the LBP dipeptide binding sequence in the 8-Cys domain, which is essential in other TGFβ-binding LTBPs [4]. LTBP2 and LTBP4 have been reported to contribute to elastogenesis by interacting with microfibrillar proteins and exhibiting overlapping functions [30]. Moreover, LTBP4 appears to inconsistently bind to latent TGFβ1, which is the only TGFβ isoform related to LTBP4. The absence of the third 8-Cys domain LAP-binding site has been detected in some LTBP4 variants [4]. Transgenic mice mutants without the LAP binding residues in LTBP4 have no remarkable phenotype, indicating that the latent TGFβ1-binding function of LTBP4 is dispensable [26].

Multiple phenotypes related to LTBP deficiency have been characterized. As with other null mutants, it is likely that the most obvious effects of loss of function have been described. Humans with loss of LTBP1 have not been reported; however, Ltbp1L^−/−^ mice died at birth because of persistent truncus arteriosus and an interrupted aortic arch [31]. LTBP2 deficiency results in primary congenital glaucoma with a defective trabecular network [32,33]. Fujikawa et al. reported that LTBP2 and LTBP4 have similar functions in the eye and that the defects observed in Ltbp2-null mice can be prevented by ectopic expression of LTBP4 [30]. LTBP3-null mice and humans share similar phenotypes, including craniofacial dysmorphism, enamel abnormalities, and acromeric and geleophysic dysplasia [34,35,36]. LTBP4 deficiency is known to be associated with various disorders (see below).

## 3. Clinical Disorders

### 3.1. Hereditary Diseases

#### 3.1.1. LTBP4-Related Cutis Laxa

Cutis laxa refers to a heterogeneous group of rare connective tissue disorders with redundant and hyperextensible skin. Autosomal recessive cutis laxa subtype C (ARCL1C) or Urban–Rifkin–Davis syndrome, caused by biallelic variants in the LTBP4 gene, is characterized by early childhood-onset pulmonary emphysema, peripheral pulmonary artery stenosis, inguinal hernias, and hollow-organ diverticula. This severe and variable disorder has, for example, been reported in 22 individuals from 18 families [37,38,39,40]. Moreover, vascular tortuosity, aneurysm, gastrointestinal diverticulosis, renal dysplasia, and bladder diverticulosis have been reported in patients with ARCL1C [37,38,40]. Disruptions in microfibril and elastic-fiber homeostasis and assembly may lead to various pathological conditions and clinical phenotypes. Regarding its molecular pathogenesis, LTBP4 interacts with and sequesters TGFβ1 to regulate its activation. However, mice with mutant variants that prevent the binding of TGFβ1 to LTBP4 show normal phenotypes, indicating that this function may require further verification [26]. Furthermore, studies have shown that LTBP4 promotes elastogenesis by incorporating with fibulin-4 [27], fibulin-5 [25,26], and fibrillin-1 [24]. In ARCL1C, elastin is inappropriately present in large globular aggregation rather than localizing in microfibril bundles and elastic fibers are not aligned well and are present in reduced amounts, indicating that impaired LTBP4 function affects elastogenesis [37,41].

In addition, LTBP4 was reported to stabilize the TGFβ receptors TGFBR1 and TGFBR2 on cell membranes [39]. LTBP4 deficiency results in reduced TGFβ signaling in dermal fibroblasts and mouse tissues, and the TGFβ receptor complex is degraded rapidly without adequate LTBP4.

As mentioned previously, ARCL1C is characterized by developmental pulmonary emphysema with impaired terminal air sac septation [37,42]. The precise mechanism of how TGFβ pathways cause LTBP4-related emphysema remains unclear, although upregulated TGFβ signaling has been observed in Lbpt4^−/−^ mouse embryos [43]. Furthermore, reduced angiogenesis, abnormal elastogenesis, altered profibrotic changes, increased myofibroblast counts, and enhanced TGFβ activity were observed in Lbpt4^−/−^ lungs; thus, these factors may play a role in the pathophysiology of LTBP4-related pulmonary emphysema [44].

#### 3.1.2. Neuromuscular Disorders

Duchenne muscular dystrophy (DMD) is caused by mutations leading to complete or near-complete dystrophin defects in the skeletal muscle [45], resulting in muscular fibro-fatty degeneration. LTBP4 has been reported to serve as a modifier in DMD, the most severe form of dystrophinopathy [46]. The LTBP4 protein is homologous in mice and humans and is highly expressed in smooth and skeletal muscles [3,4,47]. In humans, a haplotype of four missense single-nucleotide polymorphisms (SNPs) encoding LTBP4 was reported as an attractive candidate modifier of human DMD, tested in the UDP susceptibility rs2303729 (V194I), rs1131620 (T787A), rs1051303 (T820A), and rs10880 (T1140M). The two major haplotypes, which account for more than 80% of alleles in most human populations, are VTTT and IAAM [48]. Each of the four SNPs in the haplotype was found to be independently associated with later loss of independent ambulation in a recessive model, although the strongest modifier effect was associated with the full IAAM haplotype and correlated with a delay in the loss of independent ambulation at around 1.5–2 years. The most significant SNP association was identified at rs10880 (T1140M), which is near the cysteine-rich domain and has been implicated in TGFβ binding [4]. In vitro, confluent fibroblasts with the homozygote VTTT, heterozygote, and homozygote IAAM genotypes expressed equal levels of LTBP4. However, the IAAM allele was associated with reduced TGFβ/SMAD signaling. Enhanced sequestration of TGFβ in LAP by IAAM isoprotein, which is likely to be resistant to proteolysis and/or binds TGFβ, showed augmented avidity and protective effect in DMD [49].

Compared with the disrupted linkage disequilibrium and more frequent minor haplotypes observed in African Americans, European Americans present with strong linkage disequilibrium and a distinct distribution between the major VTTT and IAAM haplotypes [49]. Certainly, the LTBP4 locus can vary between populations of different ancestries. Furthermore, regarding the modifier effect of different variants, the coinherited VTTT/IAAM diplotype and other coding or regulatory variants are involved and show different degrees of linkage disequilibrium [46].

LTBP4 weakens plasma membrane stability and enhances tissue fibrosis, thereby linking the instability of sarcolemma directly to fibrotic processes. In DMD, proteins coding SNPs in LTBP4 are associated with prolonged ambulation [50].

### 3.2. Acquired Complex Diseases

#### 3.2.1. Fibrosis-Related Disorders

Scleroderma is a typical connective tissue disease involving systemic fibrosis, including cutaneous and visceral fibrosis with excessive collagen in the ECM and enhanced TGFβ activity. Lu et al. reported that LTBP4 may impact fibrotic progression in scleroderma and suggested that plasma LTBP4 is a useful clinical biomarker for diagnosing this disease [51].

#### 3.2.2. Cancers

Changes in the ECM play an important role in carcinogenesis [52], whereas TGFβ complex signaling is associated with cancer progression [5]. In addition, LTBPs serve as chaperones and transporters of TGFβ to regulate TGFβ secretion from cells [24], and incorporation of LTBPs into the ECM is crucial for the storage and activation of the latent TGFβ complex [53]. Thus, evidence shows that the regulation of LTBPs has a potential role in carcinogenesis due to their important role in ECM homeostasis and their impact on epithelial–mesenchymal transition (EMT) related to TGFβ activation [54,55,56].

LTBP4 is reportedly downregulated in human and murine ductal carcinomas in situ, invasive mammary carcinoma [57,58], and canine mammary carcinomas [59]. Promoter methylation may induce downregulation of LTBP4 in esophageal cancer [60], suggesting that it can be used as a candidate gene in a biomarker panel for neoplasias, particularly during the early stages of malignancy. In adenocarcinoma and squamous cell carcinoma cell lines, demethylation treatment leads to upregulation of LTBP4 expression, resulting in reduced cancer cell migration. Additionally, TGFβ1 plays an important role in inducing epithelial–mesenchymal transition in cancer cells and stimulates immune suppression signals as well as premetastatic niche generation [61,62]. For example, one study reported that LTBP4 suppresses metastasis in hepatocellular carcinoma [61]. In non-small-cell lung cancer (NSCLC), LTBP4 rs3786527 G>A was significantly related to lower all-cause mortality [63]. In addition, chromosome 19 aberration is quite common in lung cancer, and some genomic studies have revealed that variations in chromosome 19 lead to oncogenesis in NSCLC [56]. LTBP4 and TGFβ are located at 19q.13.2, and both are upregulated in lung cancer. TGFβ1-induced epithelial–mesenchymal transition may be involved in the pathogenesis of NSCLC [55].

#### 3.2.3. Pulmonary Disorders

In patients with LTBP4-related cutis laxa, emphysema is a dominant phenotype observed as congenital or early onset and can be progressive. These clinical manifestations demonstrate the importance of LTBP4 in lung development. LTBP4 deficiency leads to abnormal elastogenesis and elevated TGFβ levels in the lung [43]. One study showed that LTBP4 was essential to lung remodeling and injury repair, via Pdgfrβ signaling in mice [64]. Expression of LTBP4 is crucial for the development and maintenance of the pulmonary architecture, whereas its variants are associated with impaired alveolarization and airway collapse [37]. Moreover, in bleomycin-treated Thy-1-knockout mice, LTBP4 was found to regulate TGFβ activity and SMAD3 phosphorylation in pulmonary fibroblasts [65]. Therefore, LTBP4 plays a crucial role in TGFβ1 activation in bleomycin-induced lung fibrosis.

Remarkably, upregulated TGFβ and LTBP4 have been detected in the airway basal cells of smokers, and polymorphisms in genes encoding both of these proteins are associated with the susceptibility to chronic obstructive pulmonary disease. In a genome-wide association study [66], genetic variability was shown to play an important role in the development of chronic obstructive pulmonary disease. Moreover, LTBP4 significantly impacts the functional capacity of the lungs, including exercise capacity and tolerance, pulmonary function tests, and respiratory manifestation [67].

#### 3.2.4. Cardiovascular Disorders

TGFβ signaling plays a crucial role in the formation of ascending thoracic aortic aneurysm, including in patients with a tricuspid aortic valve and a bicuspid aortic valve (BAV) [68]. People with a BAV have a high risk of complications, including progressive dilation and aortic dissection and rupture. LTBP4 was found to be significantly downregulated 10-fold in the BAV group compared to the tricuspid aortic valve group. In BAV aneurismal tissue, an augmented fraction of the free small latent complex may be associated with reduced activity of LTBP4 [68].

Moreover, TGFβ1 is crucial in abdominal aortic aneurysm (AAA), and five SNPs in LTBP4 were shown to be related to a remarkable decrease in AAA growth in a UK cohort [69,70]. However, meta-analysis of AAA over 45 mm suggested that this subgroup is clinically relevant, and the LTBP4 gene should be further evaluated in genetic and functional studies of AAA disease. The LTBP4 gene may contribute to AAA progression [69].

One study of left ventricular dysfunction after acute myocardial infarction reported a panel of three genes (TNXB, TGFBR1, and LTBP4) that could potentially improve the prediction of post-myocardial infarction heart failure [71]. This result may support identification of not only a biological signature for patients with acute myocardial infarction who are at high risk of developing left ventricular dysfunction but also a novel therapeutic intervention [72]. Additionally, ADAMTS7, an extracellular metalloprotease, may cleave LTBP4 in cardiovascular system cells, potentially affecting elastogenesis [73]. Small-molecular inhibitors of ADAMTS7 play a potential therapeutic role in coronary artery disease [74], suggesting that preventing the cleavage of LTBP4 may preserve the normal ECM and help in cardiac protection.

## 4. Concluding Remarks

We described the molecular functions of LTBP4 and its role in various diseases. TGFβ-related signaling and non-TGFβ-related features play essential roles in various pathophysiologies related to LTBP4. The mechanisms underlying these diseases are not completely understood, and thus the mechanistic roles of LTBP4 should be further investigated to determine the complex associations among LTBP4, the ECM, TGFβ signaling, and non-TGFβ-related controls. These studies may lead to the development of therapeutic agents for different diseases.

## Figures and Tables

**Figure 1 genes-12-00795-f001:**
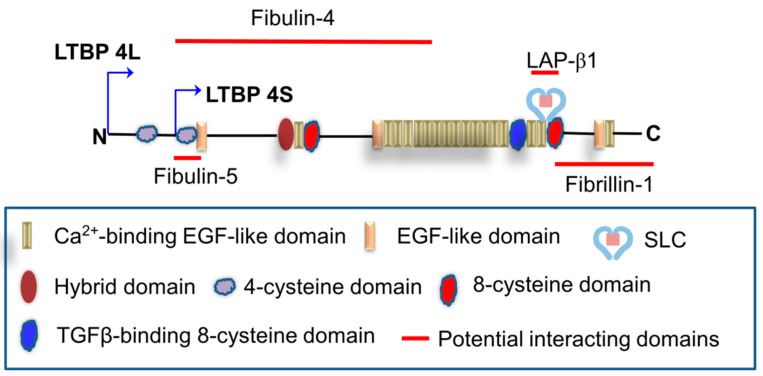
Potential binding between LTBP4 and matrix proteins. EGF: Epidermal growth factor; SLC: Small latent complex; LAP: Latency-associated peptide.

## Data Availability

Not applicable.

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
