# Peer review of "LTBP4 in Health and Disease"

_genes, 2021, doi:10.3390/genes12060795_

Round 1

Reviewer 1 Report

1.) Page 1, line 29, typo, it should read incorporate; page 1, line 39, typo, it should read LTBP4(?); page 1, line 43, typo, it should read fibrillins; line 44, there is something wrong with the sentence, potentially a doubling, please check

2.) Page 2, line 93: I would recommend not starting a new paragraph after only one sentence to make it easier to read.

3.) Page 3, line 107: LTBP4 actually interacts with a fibulin-5-fibulin-4-tropoelastin complex and seems to be required for the proper linear deposition on the microfibril network. Bultmann-Mellin et al., 2015 (Disease models & mechanisms) have shown that LTBP4 binds fibulin-4.

4.) Page 3, line 138/139: LTBP4 is involved in elastogenesis due to its interaction with fibulin-4 (Bultmann-Mellin et al., 2015), fibulin-5 (Noda et al., 2013) and fibrillin-1 (Isogai et al., 2003). Current model of elastogenesis, see for instance “Elastic fibers during aging and disease” (Heinz, 2021) or for instance Robertson et al., 2016 for the interactions between LTBP4 and other matrix proteins.

5.) Page 3, line 143: one full stop too much, needs to be removed

6.) Page 3, line 144: The authors should consider having a separate section on LTPB4 deficiency-related emphysema or is this discussed a symptom of LTBP4-related cutis laxa? In the latter case, it should be made clearer that this is a symptom of the Urban-Rifkin-Davis syndrome.

7.) Page 3, section 3.1.: It should also be mentioned that another name for the LTBP4-related cutis laxa is autosomal recessive cutis laxa 1C (ARCL1C) as there are a variety of other genetic disorders related to cutis laxa, for example two ARCLs (1A and 1B) related to mutations in the fibulin-5 and fibulin-4 genes, respectively.

Also, in this section the functions of LTBP4 should be better connected to the symptoms of the ARCL 1C. The authors mention something about LTBP4 and elastogenesis but not how this could related to the actual disease ARCL 1C.

8.) Page 4, line 158 and following lines: partly redundant information in the second sentence of section 3.3. In this section, the authors should make the connection between cancer and LTBP4 clearer. At the moment, the authors are only listing functions of LTBP4 and TGFb without establishing a connection to cancer. Is there anything known or mentioned in the cited articles as to how LTBP4 plays a role in cancer?

9.) Page 5, line 243: typo, it should read LTBP4 gene instead of LBP4 gene.

10.) General comment: A figure of some kind would be nice, for instance of the structures of the LTBP4 isoforms and potentially interactions with TGFb1 and fibrillins to put things in perspective and make them better tangible.

11.) General comment: certain abbreviations, for instance TGF(b), are not introduced. Please check all abbreviations are introduced once before using them.

Reviewer 2 Report

This manuscript reviews what is know about the structure, function and binding of LTBP4. It also summarizes all the clinical disorders associated with germline LTBP4 variants and expression variations associated with carcinogenesis and cardiovascular disease.

Overall, this is a well-written, interesting manuscript. I have some minor suggestions.

  • In section 2.4, it would be helpful to have a paragraph on the clinical disorders associated with mutations in LTBP1, LTBP2 and LTBP3 as there is some phenotypic overlap with the LTBP4 phenotypes which would be of interest.
  • The clinical disorders section intersperses conditions caused by germline predisposing variants and expression studies in carcinogenesis and cardiovascular disorders. I would suggest discussing the germline conditions first and having a separate section for tissue-specific expression studies.

On an editing note, it feels like there is a reference missing at the end of section 2.3. Also, the sentence on line 147: “However, lung development was normal” could perhaps be reworded for clarity, perhaps if the second clause of the sentence began with “despite the fact that LTBP4 was unable to bind TGFB1”.

Round 2

Reviewer 1 Report

line 44: typo, it should read glycoprotein

line 110: it should read fibulin-4

Figure 1: the reviewer suggests renaming it "Potential binding between LTBP4 and matrix proteins". In the figure it should also read Fibulin-4 and Fibulin-5 (hyphenated) as in the text.

line 139: it should read "see below" (small letter s)

line 153 and other places: sometimes the references are shown in italic and sometimes not, please check that this is consistent all throughout the document

line 161: remove the word "to"

line 140-144:  the reviewer suggests moving that part (lines 140-144) to line 129 (and the following) as it also contains structural information that is found in the first paragraph of section 2.4 between lines 118-129.

line 153: has it only be reported in 22 individuals? If not than the authors should add something like "it has for example been reported in 22 individuals..."

line 167: the reviewer suggests starting the section with: As mentioned previously, ARCL1C is characterized... (as pulmonary emphysema has also been mentioned further up)

lines 173-175: the symptoms should be moved up to where other symptoms are mentioned, i.e. after line 151/152.

line 175-177: this sentence should be removed completely, as this is already mentioned in line 161
